# The Language of Dreams: Application of Linguistics-Based Approaches for the Automated Analysis of Dream Experiences

**Valentina Elce** , **Giacomo Handjaras** and **Giulio Bernardi** *

MoMiLab Research Unit, IMT School for Advanced Studies Lucca, 55100 Lucca, Italy;
valentina.elce@imtlucca.it (V.E.); giacomo.handjaras@imtlucca.it (G.H.)
* Correspondence: giulio.bernardi@imtlucca.it

**Abstract:** The study of dreams represents a crucial intersection between philosophical, psychological, neuroscientific, and clinical interests. Importantly, one of the main sources of insight into dreaming activity are the (oral or written) reports provided by dreamers upon awakening from their sleep. Classically, two main types of information are commonly extracted from dream reports: structural and semantic, content-related information. Extracted structural information is typically limited to the simple count of words or sentences in a report. Instead, content analysis usually relies on quantitative scores assigned by two or more (blind) human operators through the use of predefined coding systems. Within this review, we will show that methods borrowed from the field of linguistic analysis, such as graph analysis, dictionary-based content analysis, and distributional semantics approaches, could be used to complement and, in many cases, replace classical measures and scales for the quantitative structural and semantic assessment of dream reports. Importantly, these methods allow the direct (operator-independent) extraction of quantitative information from language data, hence enabling a fully objective and reproducible analysis of conscious experiences occurring during human sleep. Most importantly, these approaches can be partially or fully automatized and may thus be easily applied to the analysis of large datasets.

**Keywords:** dreaming; graph analysis; word embedding; semantics

## 1. Introduction

Every night, when we fall asleep, we cease to experience the world around us. Yet, during all stages of sleep, we may be exposed to a wide gamut of internally generated experiences (i.e., dreams), ranging from simple abstract thoughts to complex movie-like narratives. With rare exceptions, these experiences develop independently from external stimuli and are not subject to voluntary control [1]. Instead, they largely draw on previously acquired memories and beliefs, and thus typically present relevant aspects of continuity with thoughts, concerns, and salient experiences of our waking self [2,3]. In light of this, dreams are thought to represent an important window on—and to potentially have a direct role in—sleep-dependent processes involving learning and memory consolidation [4,5]. Moreover, they have a tight relationship with psychophysical health. In fact, alterations in the frequency or content of oneiric experiences may accompany, or even precede, the waking manifestation of clinical symptoms related to psychiatric and neurological disorders [6–8]. From a different perspective, dreams also constitute a fundamental model for the study of human consciousness due to their nature of subjective experiences spontaneously generated by the brain independently from sensory input, motor output, and volitional processes [1]. It is thus clear that the study of dreams represents a crucial intersection between philosophical, psychological, neuroscientific, and clinical interests.

Like any phenomenal experience, dreams are subjective, 'private' experiences. Therefore, while preliminary investigations showed that it might be possible—in principle—to decode the content of dream consciousness directly from brain activity patterns [9,10], the

principal and most reliable source of insight into a dream's contents is the report provided by the dreamer upon awakening from their sleep. In particular, previous research mainly relied on two distinct approaches for the investigation of dreaming activity: the collection of free (oral or written) reports and the administration of structured questionnaires [11,12]. These methods have different and partially complementary advantages and disadvantages. For instance, questionnaires allow the attention of study participants to be directed to specific aspects of their experiences and to immediately transform qualitative information into quantitative or categorical data according to subjective estimates. At the same time, they have important limitations (e.g., [11]). In fact, they necessarily include only a limited set of questions providing a rough and incomplete description of the dream. In addition, the participants might misunderstand some of the questions or be influenced in their responses by the experimenters (experimenter bias) or the questionnaire's specific design (e.g., order or wording of questions). On the other hand, the use of free reports may allow for a more comprehensive description of oneiric experiences, including information about their narrative and logical structure. The task of providing free reports is also easier to understand for the volunteers, and they can complete it autonomously in either written or oral form. For this reason, the collection of daily free reports over a period of one to multiple weeks (dream diary) became the dominant approach for the study of oneiric experiences in the naturalistic conditions of the participants' home environment. Crucially, this particular experimental approach has a special role in dream research, as previous studies showed that dreams collected in the sleep laboratory typically differ from those reported upon awakening in non-laboratory conditions. For instance, oneiric experiences collected upon awakenings during laboratory experiments are typically poorer than home-based reports (e.g., fewer emotions) and may incorporate elements and themes of the experimental setting [13,14]. Nevertheless, the use of free reports for the analysis of dream experiences also entails important limitations and methodological issues. These include, for instance, possible inter-subject differences—such as those in linguistic skills and verbosity—that could affect how dreams are described rather than actually experienced. However, the most important and well-known issue concerns the objective and unbiased extraction of quantitative information from individual reports [15].

Two main types of information are commonly extracted from free dream reports: structural and semantic (content-related) information [15]. The most typical examples of structural information are the total word count (i.e., the total number of words used for describing the oneiric experience) and its variants, such as the number of sentences or text lines [16,17]. Importantly, while these parameters are easy to compute and convert into quantitative objective values, they only provide a very rough estimate of the reports' actual structure and are likely to be greatly affected by inter-individual differences in verbosity and linguistic skills [16,18]. The analysis of dream contents commonly relies instead on quantitative 'scores' assigned by two or more (blind) human operators through the use of arbitrary scales, such as the scale by Hauri and colleagues [19], the rating system developed by Schredl [20], and the Hall and Van de Castle coding system [21,22]. In particular, the Hall and Van de Castle allows the scorer to identify ten categories of elements appearing in dream reports, that are: characters, interactions, emotions, activities, striving, (mis)fortunes, settings and objects, descriptive elements, food and eating, elements from the past. The main limitation of these approaches lies in their complexity and dependence on trained human raters. In fact, the scoring procedure is highly time-consuming and thus difficult or impossible to apply for the analysis of very large datasets. Moreover, reproducibility of results may be affected by the level of inter-rater agreement within and across studies, which may vary as a function of the raters' training or for distinct semantic categories [15].

Crucially, while still relying on blind raters' judgement, several studies also tried to model the language of dreams by exploring the occurrence in dream reports of specific linguistic features, such as the frequency of words referring to visual imagery (i.e., visual nouns, actions, modifiers and spatial relations) and auditory imagery (i.e., explicit and implicit speech [23]) or the degree of grammatical reference to past and future events [24].

These studies represent a prolific attempt to define quantitative and objective measures for the analysis of dream reports according to specific features and criteria that are less susceptible to raters' personal interpretation. However, such approaches share with classical coding systems most limitations derived from full reliance on human raters. Therefore, starting with the seminal work by Schwarz [25], further studies tried to address this issue by resorting to the most advanced techniques of computational linguistic analysis, which are acquiring greater functionality and relevance through the implementation of new instruments and methodological approaches.

Within this review, we will show that methods borrowed from the field of linguistic analysis could be used to complement and, in some cases, replace classical measures and scales for the quantitative structural and semantic assessment of dream reports. In fact, these methods may allow the extraction of quantitative information from language data, hence enabling a more objective, reproducible and less operator-dependent analysis of verbal reports describing conscious experiences occurring during human sleep. Most importantly, these approaches can be partially or fully automatized and may thus be easily applied to the analysis of large databases.

## 2. Graph Analysis: Exploring the Structural Properties of Mentation Reports

### 2.1. Overview

Graph analysis is a common framework for evaluating the type, strength, and direction of relationships between elements using a graph representation in which data elements correspond to nodes and relationships correspond to edges [26]. This approach allows the researcher to examine the overall structure of a graph, quantifying the pairwise relationships among its units and evidencing features that could not be captured by considering only its elements/nodes. Hence, when applied to language processing, graph analysis determines the non-semantic word-to-word structural organization of speech [27]. While this method could be applied to both written and spoken dream reports, previous studies mainly used it on oral speech samples. Once speech is represented as a graph, it is possible to calculate several mathematical attributes that quantify local and global topological characteristics (i.e., regarding the arrangement of nodes and edges; [28]). In this respect, graph analysis of dream reports may provide information that extends and complements the classical total word count and related measures. Moreover, as detailed below (Section 2.2), while some connectivity attributes may depend on—and thus present strong collinearity with—word count, graph analysis offers different methods to account for this confound, thus deriving meaningful structural properties independent of report length [29].

### 2.2. Description of the Approach

In practical terms, graph theory represents free speech as a trajectory where each word is defined as a node, and the temporal sequence between every consecutive pair of words is defined by directed or undirected edges [28]. The graphs can be generated by means of automatic software, such as the custom-made Java software SpeechGraph, developed by Mota and colleagues [28]. Of note, several automated and/or manual preprocessing steps should be applied prior to the creation of a graph in order to obtain a meaningful and interpretable representation of the analyzed speech sample. However, as detailed below (Section 2.4), there is currently no agreement among published studies regarding a specific pipeline.

In mathematical terms, each (directed or undirected) graph is represented by an adjacency matrix, in which rows and columns are assigned to the nodes, and the presence of an edge is symbolized by a numerical value. For instance, in a so-called multigraph (a graph which is permitted to have multiple edges), "0" would indicate no edges, "1" one edge, and "2" two parallel edges. The adjacency matrix representation can be used to calculate several mathematical attributes that quantify local and global topological characteristics of the graph network. In particular, most studies focused on the following

sets of parameters (see Table 1 for further descriptions): general graph attributes (e.g., number of nodes and edges), recurrence attributes (e.g., recurrent/parallel edges and loops), connectivity attributes (e.g., size of connected clusters), and global attributes of the graphs. However, other topological properties could, in principle, also be computed (e.g., see [30]).

**Table 1.** The table describes the main mathematical attributes that were computed through graph analysis in the published literature on dream experiences.

| Category | Attribute | Description |
|---|---|---|
| General graph attributes | Total number of nodes | Total number of unique words |
| | Total number of edges | Total number of connections |
| Recurrence attributes | Repeated edges | Sum of all edges linking the same pair of nodes |
| | Parallel edges | Sum of parallel edges linking the same pair of nodes |
| | Loops of one (or more) node | Sum of all edges linking a node with itself or sum of all loops containing two or more nodes |
| Connectivity attributes | Largest connected component (LCC) | Number of nodes in the maximal subgraph in which each pair of nodes is directly linked by an edge |
| | Largest strongly connected component (LSC) | Number of nodes in the in the maximal subgraph in which each pair of nodes has a mutually reachable path |
| General graph attributes | Average total degree | Given a node n, the sum of "in" and "out" edges represent its total degree; average total degree is the sum of total degree of all nodes divided by the number of nodes. |
| | Density | Number of edges (E) divided by the number of possible edges, according to the total number of nodes (N): $(D = 2 \times E/N \times (N-1))$. |
| | Diameter | Length of the longest shortest path between the node pairs of a network. |
| | Average shortest path | Average length (number of steps along edges) of the shortest path between pairs of nodes of a network. |
| | Clustering coefficient | The set of fractions of all node neighbors that are also neighbors of each other. |

As mentioned above, connectivity attributes may show important collinearity with total word count, which may thus represent a possible confounding factor [29]. Two different strategies were applied in the literature to account for this issue: sliding word windows and normalization against random graphs [31]. In the first case, graph attributes are separately computed for overlapping windows with a fixed number of words (e.g., 30 words per window and 1-word steps between subsequent windows). Then, mean values may be computed across all windows for graph measures of interest. In the other approach, multiple random graphs (e.g., 1000) are generated by shuffling the words in each report, and graph attributes of interest are calculated at each iteration. Then, values obtained from the original (non-shuffled) graph are normalized against the average of the metrics obtained from the random graphs (e.g., ratio). Of note, this latter approach cannot be applied to features that remain equal in the original and random graphs, such as the number of nodes and edges.

### 2.3. Applications in the Field of Dream Research

Graph analysis applied to the analysis of dream reports was employed in studies on healthy subjects and for investigating changes in dream features within clinical populations (Table 2). In healthy subjects, graph analysis of mentation reports was used to investigate potential stage-specific structural properties. Specifically, Martin and colleagues investigated both differences in graph structure between REM and NREM dreams and the relationship between non-semantic graph attributes and dream report length, defined here as the total number of words used to describe a dreaming experience after excluding redundancies, repetitions, interjections, corrections and dreamer's comments about the mentation [27]. By analyzing 133 reports collected from 20 participants in controlled labo-

ratory awakenings from REM and N2 sleep, the authors found that REM dream reports were characterized by higher word count and graph connectedness (expressed in terms of LCC and LSC; see Table 1) as compared to N2 dreams. A sliding-window approach was applied to account for differences in report length across stages and participants (see Section 2.2). However, when dream reports were compared to randomly generated graphs, the authors did not find any difference between REM and N2 dreams, thus suggesting that, irrespective of the sleep stage they were conceived in, the structure of healthy subjects' dream reports did not approximate to random speech, as it might be instead in pathological conditions ([32], see below). Finally, in order to understand whether measures of report length and graph connectedness might predict the degree of dream complexity, verbal reports were scored according to the Perception-Interaction Rating Scale (PIRS, [33]), an ordinary scale that rates the level of interaction between dream characters and dream environment. The authors found that higher PIRS scores of dream report complexity corresponded to increases in both dream report length and graph connectedness and to a decrease in graph random-likeness.

**Table 2.** Overview of studies using graph analysis to assess structural properties of dream reports (described in Section 2.2).

| First Author | Year | Population(s) | Analysis | Indices | Main Results |
|---|---|---|---|---|---|
| Mota, NB | 2012 | 8 schizophrenic patients; 8 manic patients; 8 healthy controls | Psychiatric populations' vs. healthy controls' dream reports | General graph attributes; Recurrence attributes; Connectivity attributes; Global attributes | Increased number of waking nodes and parallel edges in manics, reduced connectivity in schizophrenics |
| Mota, NB | 2014 | 20 patients with Schizophrenia; 20 patients with Bipolar Disorder; 20 healthy controls. | Psychiatric populations' vs. healthy controls' dream reports | General graph attributes; Recurrence attributes; Connectivity attributes; Global attributes | Random-like connectedness more prevalent among schizophrenic patients |
| Mota, NB | 2016 | 25 patients with Schizophrenia; 20 patients with Bipolar Disorder; 28 healthy controls | Psychotic lucid dreamers vs. non psychotic lucid dreamers; Lucid dreamers with psychotic symptoms vs. non lucid dreamers with psychotic symptoms | General graph attributes; Recurrence attributes; Connectivity attributes; Global attributes | Smaller clustering coefficient in schizophrenia lucid dreamers relative to non-lucid dreamers |
| Mota, NB | 2017 | 21 subjects with recent-onset psychosis; 21 healthy controls | Subjects with recent-onset psychosis' oral reports vs. healthy controls' oral reports | Graph connectedness measures: Total number of edges; Largest connected component; Largest strongly connected component | Reduced connectivity in schizophrenia and bipolar disorder patients |
| Martin, JM | 2020 | 20 healthy subjects | REM vs. NREM dream reports | Graph connectedness measures: Largest connected component; Largest strongly connected component | Increased graph connectedness in REM dream reports as compared to NREM dream reports |
| Mota, NB | 2021 | 67 healthy subjects | Non-pandemic vs. pandemic dreaming | Graph connectedness measures: Largest connected component; Largest strongly connected component | No difference in graph connectedness between pandemic and non-pandemic dreams |

Graph analysis of dream reports found extensive application in the study of possible differences between clinical populations and healthy individuals. This interest was motivated by the fact that speech disorganization characterizes in particular formal thought disorder (FTD), that is "any impairment in the production of language and subjective alteration in the thought process" [34], therefore affecting the form rather than the content of thinking. FTD is a core symptom of several neuropsychiatric disorders, such as schizophrenia, major depressive disorders, and mania and, from a linguistic point of view, is characterized by simpler syntax, poverty of speech and content, incoherence, and impaired prosody. Therefore, several studies investigated whether increased speech disorganization could be found in dream reports of psychiatric populations and used to distinguish

patients from healthy controls. For instance, Mota and colleagues [29] initially compared reports of recent dreams provided by psychotic patients, including 8 schizophrenic patients and 8 manic patients, with those of healthy volunteers. Each report was represented as a graph to calculate several attributes that were eventually normalized by the number of words in the report. The authors found that manic dream reports were characterized by a greater number of parallel edges, more waking nodes (i.e., nodes used to describe waking rather than dreaming events), and denser graphs (smaller diameter and average shortest path) as compared to schizophrenic patients. These observations are consistent with the symptomatology of mania, which may include logorrhea and flight of thoughts. Instead, schizophrenic patients showed more nodes and a higher average total degree per word than manics, indicating that schizophrenics tended to address each topic only once, with minimal redundancy or repetition of concepts.

In a subsequent work, Mota and colleagues [28] measured speech graph attributes in a sample of waking and dreaming experience reports recorded by patients affected by schizophrenia, bipolar disorder type I patients and non-psychotic controls. By analyzing graph attributes of participants' reports computed using a sliding window approach, the authors were able to automatically sort schizophrenia and bipolar patients undergoing psychosis, and also to separate these psychotic patients from subjects without psychosis. Interestingly, bipolar patients and control subjects had indistinguishable by waking reports, while the former showed significantly less connectivity in dream reports (reduced LCC and LSC). Moreover, in line with previous work, both waking and dream reports produced by schizophrenic patients were characterized by smaller and less connected graphs relative to control and bipolar subjects. Interestingly, the authors also found that dream graph connectivity attributes negatively correlated with clinical symptoms measured by psychometric scales.

Later, the same authors further investigated the relationship between graph connectivity attributes and schizophrenia in order to verify whether connectivity measures of speech disorganization could allow for the classification of the severity of negative symptoms [32]. Within this study, they collected oral reports from 21 patients undergoing the first clinical interview for recent-onset psychosis and appropriately matched controls. After a 6 months follow-up, 10 patients were diagnosed with bipolar disorder and 11 with schizophrenia. During the first clinical interview, the subjects were asked to report a dream, the oldest memory they could recall at the moment of the interview and to talk about their previous day, for a maximum of 30 s per each report. Moreover, participants were presented with a highly positive image, a highly negative image, and a neutral image, and they were asked to describe an imaginary story based on them. The authors found that reports produced by schizophrenic patients regarding negative images had a less connected structure (i.e., fewer edges and smaller LCC and LSC) and were more similar to random graphs as compared to those produced by the other samples. Such an effect also persisted once participants' verbosity was controlled by dividing speech graph attributes by word rate, that is the amount of words produced in the 30 s reports. Crucially, a linear combination of connectedness attributes reached high classification accuracy for negative symptom severity and of schizophrenia diagnosis 6 months in advance.

Dreaming in psychotic populations was further explored by Mota and colleagues [35] through the study of lucid dreaming, representing a peculiar state in which the dreamer becomes aware of being in a dream [36]. Interest toward lucid dreaming was especially motivated by the fact that cortical areas activated during this state largely overlap with brain regions that are impaired in psychotic patients who lack insight into their pathological condition [37]. Specifically, the authors explored lucid dream features and psychiatric symptoms in a cohort of 45 patients with psychotic symptoms (25 with schizophrenia and 20 with bipolar disorder), which were compared to a sample of 28 healthy subjects. Participants were asked to recall the most recent dream they could remember and to answer a set of questions regarding lucid dreaming. Interestingly, psychotic patients reported a similar incidence of lucid dreaming as control volunteers, and no differences

in symptomatology were found between lucid and non-lucid dreamers among psychotic patients. Consistent with this, graph analysis revealed only minor differences between these groups. In particular, it was found that dream reports produced by schizophrenic lucid dreamers were represented by less complex speech graphs (i.e., smaller clustering coefficient) in comparison to reports of non-lucid dreamers.

In the context of the recent interest toward the effects of the COVID-19 pandemic on sleep and dreams, Mota and colleagues also applied graph analysis to investigate potential changes in the structure of dream reports of healthy individuals due to mental suffering and stress during the lockdown period [38]. Specifically, they analyzed and compared structural and semantic properties of dream reports collected from 67 individuals before the outbreak of COVID-19 with those collected after the spread of the virus and the adoption of precautionary measures such as self-isolation and social distancing. In this case, while from a semantic point of view (see Section 3.3) the study evidenced that pandemic dreams had a higher proportion of sadness and anger-related words and semantic similarities to contamination and cleanness, from a structural point of view, the authors did not find any change in dream report connectedness when accounting for the number of words using a sliding window technique.

To sum up, these studies provided preliminary evidence for the potential value of graph analysis in the study of dream experiences under physiological and pathological conditions. Interestingly, they show that metrics derived from graph analysis may allow one to study the structure of dream reports above and beyond the classical measure of total word count, with the additional possibility to distinguish between actual structural changes and changes that simply reflect verbosity differences.

### 2.4. Methodological Considerations

The information that graph analysis can provide strongly depends on the intrinsic characteristics of the text, which may, in turn, be affected by the methodologies chosen to both acquire and preprocess the dream reports.

With respect to data acquisition, dream reports could be collected in written or oral form. Both strategies have advantages and disadvantages, and it is up to the researcher to choose the most suitable approach in relation to the study hypothesis. In particular, the collection of written reports upon awakening is a more demanding task, as compared to the recording of oral reports. However, it provides more cohesive and shorter reports [39], characterized by a tendency to concentrate meaning in nouns and to reduce the number of propositions. Written reports exhibit a higher lexical density, that is the ratio of lexical items (i.e., content words such as nouns, verbs, adjectives and adverbs) to the total number of words [16]. Instead, the acquisition of oral dream reports is simpler and allows for spontaneous textual samples that can be revised neither by the speaker nor the experimenter and that may thus often contain false starts, repetitions, and digressions (indeed, the average total word count observed for written reports was ~80 words, vs. ~150 words for oral reports) [16]. The use of oral reports requires the transcription of a large number of recordings, which could be performed manually or automatically. Of note, despite being faster, automated procedures may lead to more frequent transcription errors, thus adding noise to the data. For this reason, an operator-dependent inspection of the reports may be performed to correct potential errors. However, the choice of a particular strategy is necessarily influenced by experimental factors such as the size of the dataset. Importantly, the transcription of oral dream reports also typically involves the inclusion of punctuation marks, which may be added automatically, by the transcription software, or manually. This operation could have a relevant impact on the analysis results depending on the criteria applied to add the punctuation and to whether or not two words separated by a particular punctuation mark will be considered as linked. As specified below, though, most published studies removed and thus ignored all punctuation marks and symbols during the preprocessing phase.

In discussing the difference between oral and written dream reports, it is also important to point out that the former might show peculiar structural features that are usually not found in written dream diaries [40], such as structurally well-formed statements that are illogical or nonsensical in meaning, ill-formed linguistic structures and sentences, filler words (e.g., "anyway", "basically"), repetitions (e.g., "I dreamt of a dog of a dog that"), fragments of words (e.g., "I have dre I have dreamt"), and interjections (e.g., "hmm", "yeah", "ahem"). Moreover, verbal dream reports are more likely to contain corrections of previous statements (e.g., "actually it was blue, not red") and speakers' comments regarding the dream (e.g., "that's strange", "I don't know why"). Nevertheless, some of these elements, such as filler words and dreamers' comments about the mentation, may be also found in written dream reports. Since graph analysis represents and evaluates the word-to-word organization of verbal reports, the above linguistic elements may significantly affect graph topology and related descriptive metrics. The automatic or manual removal of one or more of the above categories may thus be considered according to the particular study hypotheses and aims. For instance, list- or rule-based approaches could be used to filter-out so-called 'stop-words', such as filler words, interjections, and short words unlikely to carry relevant meaning (e.g., words with less than three letters; [31]). Of note, the definition of linguistic items as elements of no interest represents an arbitrary step in the preprocessing stage and could potentially determine the exclusion of expressions that are actually meaningful to the dreamer. Being arbitrary, this definition might change across different studies and according to the experimental hypothesis. In this respect, it is crucial to define and describe specific rules for the identification of such elements for the purpose of reproducible results. However, the removal of elements such as speakers' comments or corrections is (at present) not automatable and must be performed manually, and preferentially by operators blind to the hypotheses and experimental conditions of interest [27].

To sum up, the typical preprocessing pipeline of textual dream reports includes several standard and optional steps. Common basic preprocessing steps include conversion of text to lowercase and tokenization, which is the process of splitting a text sample into space-separated elements, such as words, characters, or subwords. This step may be preceded by manual removal of comments or expressions that may bias subsequent analyses. Then, lemmatization could be performed to reduce the number of common words present in a text by converting each inflected word in its 'canonical form'. For example, the verb "to walk" might appear in the form of "walk", "walked", "walks" or "walking", its canonical form is "walk", also defined as lemma or dictionary form. By performing lemmatization, inflected forms of the same word will represent a single node, rather than multiple nodes, in the graph [31]. Lemmatization can be implemented by means of natural language processing tools employing lexical databases such as WordNet [41] or the Stanford CoreNLP Natural Language Processing Toolkit [42] as reference. An alternative approach to lemmatization is stemming, which is the process of reducing inflected words to their 'root' (or 'stem') form. The difference is that stemming operates using pre-defined rules on a word-by-word basis and cannot discriminate between words that have different meanings depending on the context. For instance, through stemming, the word "meeting" is always converted into "meet", while lemmatization operates depending on whether the word is recognized as a verb form or a noun according to the context. Thus, while stemming is typically faster and easier to implement with respect to lemmatization, it may have a lower accuracy. Finally, automated approaches may be applied to remove symbols, punctuation marks and stop-words.

Importantly, while the described preprocessing steps may unveil particular aspects of the dream reports' structure that would not be accessible by analyzing the raw data, it should be kept in mind that the unaware removal of specific linguistic units may strongly affect textual data and could cause a loss of relevant information. This consideration is especially important given that a direct and thorough assessment of the impact of different preprocessing strategies on the final results has not yet been performed. Given

the above premises, the preprocessing strategy should always be carefully selected (and thoroughly described) according to the research hypothesis and the particular type of data to be analyzed.

## 3. Semantic Analysis for the Study of Dream Content

### 3.1. Overview

Two main classes of methods have been used in the literature for the automated identification and analysis of dream content; distributional semantics and dictionary-based approaches (Table 3). While dictionary-based approaches explore the occurrence frequency of unique words regardless of their context, distributional semantics analyzes the relationship between each word and its context of occurrence.

**Table 3.** Overview of studies using dictionary-based methods or distributional semantics approaches (described in Section 3.2).

| First Author | Year | Dataset | Analysis | Indices | Main Results |
|---|---|---|---|---|---|
| Schwartz, S | 2004 | Corpus of 1770 author's dreams; Author's 3-week diary of real events; Corpus of 1000 reports provided by 200 undergraduate students (100 males, 100 females) | Author's dreams vs. author's diary of real events; author's dream vs. undergraduate students' dreams | Correspondence Analysis (CoA); Cluster Analysis | Dreams were structured as "self-referential fiction"; Narrative of waking experiences resembled newspapers or essays; The author's and students' dreams were grouped into few principal clusters |
| Altszyler, E | 2017 | DreamBank corpus: 19000 dreams reports from 59 subjects | LSA vs. word2vec for semantic analysis | Word embedding techniques (LSA, word2vec) | LSA outperformed word2vec in detecting semantic relatedness between dreams and waking reports |
| Fogel, SM | 2018 | 24 healthy participants; Spatial navigation condition, N = 12; Tennis condition, N = 12 | Performance in motor or spatial tasks vs. degree of task incorporation into dreaming experience | Semantic similarity measured with WordNet | Significant relationship between participants' performance in tasks and degree of incorporation into early dreams |
| Sanz, C | 2018 | Erowid corpus—Dreamjournal corpus | Dreaming experience vs. phenomenological effects of psychedelic substances | Word embedding techniques (LSA) | LSD induced hallucinatory experiences more similar to highly lucid dreams |
| Bulkeley, K | 2018 | SDDb Baseline Dreams corpora: 5208 baseline dreams; 625 reports of one's worst nightmares; 388 dream reports of lucid self-awareness. | Baseline dreams vs. lucid dreams and nightmares | Word frequency counting with LIWC | Lucid dreams had frequent references to cognitive processes and few words referring to visual perception; Nightmares had more references to anxiety, anger and sadness than baseline dreams |
| Mota, NB | 2020 | Control group (pre-pandemic dreams): 31 healthy participants; Experimental group (pandemic dreams): 31 healthy participants | Pandemic dreams vs. non-pandemic dreams | Word frequency counting with LIWC; Word embedding techniques (fastText) applied to external corpus | Pandemic dreams had the higher average semantic relatedness to the words "contamination" and "cleanness" |
| Pesonen, AK | 2020 | Dream reports collected from 811 respondents | Dream content vs. sleep quality and perceived level of stress in pandemic lockdown | Analysis of word associations | Participants with an increased stress level reported a higher frequency of nightmares; 55% of bad dream clusters were related to pandemic-specific themes |
| Mallett, R | 2021 | 54 healthy participants | Affective states in dreams vs. morning mood | Word frequency counting with LIWC | Dreams with higher degree of reference to anxiety, death, the body and first-person related to more negative morning mood; Dreams with positive emotion, leisure, ingestion and plural first-person references were associated with less negative morning mood. |

Dictionary-based methods allow for a word-by-word classification of lexical items by comparing textual data with predefined lists of lexical entries representing different

semantic categories of interest. In other words, these approaches involve counting the number of words that appear in each dream report that belong to particular categories available within the software used to perform the analysis (see below) or predefined by the experimenter based on specific aims and hypotheses. In its simplest and most common implementation, dictionary-based methods are used for so-called sentiment analysis, which aims to identify references to positive or negative affective states in text documents. However, the same approach could be used to identify and quantify—in terms of word frequency—a wide variety of contents and themes in dream reports. While the use of dictionary-based methods is limited by the necessity to find or generate predefined categorical word lists of interest, their implementation is simple and associated with low computational and time costs. Moreover, and most importantly, they can be applied to datasets of any size, in contrast to distributional semantics methods (see Section 3.4).

The distributional semantics framework allows capturing of meanings, semantic relationships, similarities between words, and the context of different words as they are naturally used by speakers within specific grammatical environments in large samples of language data. An underlying assumption of distributional semantics is that linguistic items with similar distributions have related meanings: that is, words that are used and occur in the same contexts tend to represent related (not necessarily similar) concepts [43]. This means that semantic representations of words occurring in specific textual data may be extracted by analyzing patterns of lexical co-occurrence in large language corpora. Through the use of linear algebra as a computational tool and representational framework, in fact, distributional information can be collected in high-dimensional vectors (or 'embeddings'), and the degree of semantic relatedness between two or more words can be thus quantified in terms of vector similarity [44]. In practice, different word embedding techniques may be applied to map words or phrases from a text sample to N-dimensional vectors of real numbers. The vector values for each word represent its position in a relatively low-dimensional linear space, the embedding space [45]. As described by Levy and Goldberg, word embeddings quantify attributional similarities between vocabulary items, that is the degree of correspondence between two or more words [46], in terms of shared semantic or syntactic properties (for instance, the word "dog" and the word "wolf" have a high degree of attributional similarity), and group words that share such properties in the projected space. For example, in the embedding space synonyms or words that appear in similar contexts will be found close to each other, while words with unrelated meanings will have a larger distance between them. Interestingly, the same methodologies could be applied not only to investigate the relationship between words, but also between different documents (i.e., dreams), through the creation of document embeddings. This approach may be used to investigate the possible existence of dream clusters, similar to what has been done in a previous study by Schwartz ([25]; see Section 3.3).

### 3.2. Description of the Approaches
#### 3.2.1. Dictionary-Based Text Analysis

Dictionary-based approaches rely on word frequency count for specific categories. For example, the Linguistic Inquiry and Word Count (LIWC) system, developed by Pennebaker and colleagues [47], is a text analysis software that calculates categorical word frequencies. This software analyzes language data word by word by comparing each item to a dictionary file that is structured as a collection of words defining different categories, including, among others, common linguistic features (e.g., articles and pronouns), affective processes, cognitive processes (e.g., insight, causation) and perceptual processes. Through such comparison, LIWC provides frequencies of word usage for each category within textual data. The software is available for different languages.

A distinct but related approach may rely on a predetermined lexical-semantic database to identify relationships between themes or concepts of interest. An example of such a database is the WordNet project [41]. WordNet displays a collection of more than 150,000 words along with a brief definition. Each word is also categorized in terms of part

of speech (e.g., noun, verb, adverb, etc.) and grouped into several sets of synonyms, or 'synsets'. WordNet provides a hierarchical organization of words according to specific semantic relations (e.g., synonyms, antonyms, parent-child relations). Such a hierarchy is also represented as a network of words, or 'hypernym tree'. This network may be used to find in text, and thus count, words that are similar in terms of shared semantic traits or to measure the distance (in terms of edge count) between different elements of the network.

### 3.2.2. Distributional Semantics Approaches

Two main families of word embedding approaches are commonly employed; global matrix factorization methods, such as Latent Semantic Analysis (LSA; [48]), and local context window methods, such as word2vec [49]. In both cases the input is a text corpus, and the output is a set of vectors of continuous values representing each distinct word occurring in the dataset. The cosine similarity between vectors is then typically used to quantify the degree of semantic relatedness between the words represented by the vectors [50].

The LSA approach is based on the assumption that words with similar meaning will occur in similar pieces of text, or documents [51]. A fundamental step in LSA is the creation of a word-by-document frequency matrix, which is built by compiling a list of the words occurring in the whole corpus of documents (e.g., dream reports) and computing the frequency of each word in each document. Then, a mathematical transformation (e.g., tf-idf or PPMI; [52]) may be applied to reduce the weight of uninformative high-frequency words in the words-documents matrix [53]. Finally, a mathematical technique called singular value decomposition (SVD) is used to obtain a low-dimensional vectorial representation of every word present in the corpus. This transformation has the effect of preserving the most important semantic information while reducing noise and other undesirable artifacts of the original space. In practice, each word is thus represented as a vector of values, and words that tend to occur together will be represented by similar vectors and will occupy a similar location in a multi-dimensional space.

Word2vec is an algorithm that uses a neural network model to learn word associations from a large corpus of text. Of note, word2vec operates according to one of two neural network language models, Continuous Bag of Words (CBOW) and Skip-gram [54]. These models share the use of a window of predefined length that moves along the corpus in order to train the network with the different words occurring in the window at each iteration. The CBOW model is designed to predict the word in the center of the window according to the surrounding words, which constitute its context of occurrence; conversely, the Skip-gram model is trained to predict the context of occurrence based on the central word of the window. While CBOW is faster than Skip-gram, the latter is often preferred because it provides more accurate results for infrequent words. In both cases, the output of the word2vec neural network is a matrix in which each word is represented by a vector.

Importantly, while LSA and word2vec are among the most commonly used approaches, several other popular algorithms exist that can provide vector space representations of words, including for instance fastText ([55]; see Section 3.3.2) and GloVe (Global Vectors for Word Representation; [50]). FastText is an extension of the word2vec model that regards each word as composed of character n-grams, so that the embedding for a word is derived from the sum of its character n-grams. GloVe is an unsupervised 'count-based' learning model that is trained on aggregated global word-word co-occurrence statistics from a corpus.

### 3.3. Applications in the Field of Dream Research

#### 3.3.1. Dictionary-Based Text Analysis

The use of modern dictionary-based text analysis approaches had an important precursor in studies that employed search engines of large dream databases (e.g., Dream-Bank; [56]) to identify the occurrence in dream reports of predefined word strings related to particular themes and contents of interest [56–60]. While limited in their application to

pre-existing databases, these works clearly demonstrated the potential value of automated, dictionary-based methods as fast, reliable, and objective alternatives to manual scoring.

In 2018, Bulkeley and Graves analyzed a collection of 5208 dream reports extracted from the Sleep and Dream Database (SDDB [61,62]). The reports were originally collected by asking participants to describe their most recent oneiric experience. By performing content analysis through the LIWC system, the authors found that dream reports were specifically characterized by a high frequency of words belonging to the categories "focus on the past", "personal pronouns", "motion", "space", "home", "first-person singular words", "dictionary words" and "authenticity". Moreover, in order to assess whether such an approach may allow to distinguish different kinds of dream data, the authors also compared recent (baseline) dreams to collections of nightmares and lucid dreams. The analysis revealed that lucid dreams had the highest frequency of references to cognitive processes and the lowest frequency of words referring to visual perception. Nightmares were characterized by the most references to death and by a higher percentage of references to anxiety, anger, and sadness, as compared to baseline dreams.

The LIWC software was more recently employed to explore the relationship between dreams and subsequent waking mood [63]. Within this study, dream reports were collected from a sample of 54 healthy participants who were also asked to rate the impact of each dream on their subsequent waking mood. LIWC was used to quantify the degree of linguistic reference to affective states in dream reports, and obtained estimates were then correlated to participants' morning mood. The study evidenced that dreams characterized by a higher degree of reference to "anxiety", "death", "the body" and "first-person" (i.e., singular first-person pronouns such as "I", "me", "my", "mine", "myself") related to more negative morning mood, while the opposite was true for dreams including "positive emotion", "leisure", "ingestion" and "plural first-person" (i.e., plural first-person pronouns such as "we", "us", "our", "ourselves").

LIWC was also applied to evaluate the emotional valence of dreams collected during the COVID-19 pandemic relative to dreams collected before the COVID-19 outbreak (also see Section 2.3). Here, the software was employed to measure the proportion of words referring to emotional contents in dream reports (i.e., positive/negative emotions, anxiety-, anger-, and sadness-related words). Consistent with the idea of a continuity between waking and dreaming experiences [2], the authors found that the proportion of anger- and sadness-related words was higher in pandemic dreams.

In 2018, Fogel and colleagues applied a dictionary-based approach to explore the relationship between pre-post sleep memory performance improvement and learning-related dream incorporation [64]. Within this study, participants were first trained on either a spatial navigation task or a tennis task. Subjects were then asked to mentally rehearse the task with their eyes closed and then to verbally describe the sequence. A daytime nap session followed, where participants were awakened after short periods of NREM sleep and asked to provide a dream report. During the nap session, the experimenters also collected verbal reports of subjective experiences that participants had while attempting to fall asleep, after a minimum of 10 s of wake. Finally, participants were retested on the task acquired before the nap session. The authors measured dream incorporation as the average degree of semantic similarity between participants' description of the tasks, dream and daydream reports by means of the WordNet English language lexical database [65]. Specifically, Fogel and colleagues created a list of synsets (see Section 3.2.1) corresponding to the words occurring in each verbal report. Since a given word can belong to several synsets, all synsets for each word from the corpus were included in the analysis. Then, the authors calculated the similarity between word meanings and the relationship between synsets in the hypernym tree on a scale from 0 (no semantic relationship) to 100 (completely synonymous) by computing the shortest number of edges from one synset to another within the hierarchical WordNet structure (Wu–Palmer similarity [66]). The semantic distance was computed iteratively between each pair of synsets corresponding to a given word from the wake reports and the dream reports. The same pairwise calculation was

performed for the comparison between wake reports and daydream reports. The study evidenced that participants' performance in motor or spatial tasks was related to the degree of incorporation of task-related elements into the content of dreams. Such a relationship reached statistical significance for early dream reports only, as compared to late dreams.

### 3.3.2. Distributional Semantics Approaches

Already in 2004, Schwarz applied an approach closely related to LSA to the analysis of dream and wakefulness reports with the aim of describing basic commonalities in dream activity [25]. Specifically, the author used factorial Correspondence Analysis (CoA) and cluster analysis to assess the degree of relatedness between distinct dream reports according to the frequency distribution of words in each document (i.e., starting from a word-by-document frequency matrix). Similar to LSA, CoA uses SVD, but it works on the sets of row- or column-profiles and uses the chi-square distance between pairs of row- or column-points, eventually providing the coordinates of row- or column-points in a lower-dimension space. By applying this approach on a corpus of dream and wakefulness experiences reported by the author and on a database of dreams provided by 200 undergraduate students, Schwarz found that dreams were structured as "self-referential fiction", that resembled novels or theater play, where the dreamer seemed to play the role of a first-person actor. Conversely, the narrative of waking experiences resembled newspapers or essays, particularly closer to dialogues or second person's writings. Cluster analysis evidenced in the author's corpus five main clusters of dreams, including, among others, affective/academic environment and fears/physical danger. In undergraduate students, the analysis revealed five different clusters, inclusive of reports characterized by motion, violence, vehicles and fear-related words and reports referring to sexual and family-related elements.

More than ten years later, Altszyler and colleagues [67] used LSA and word2vec approaches to analyze the usage patterns of the word "run" between wakefulness reports extracted from TASA (Touchstone Applied Science Associate) and UkWaC (Web as Corpus, UK) corpora [68,69] and dream reports extracted from the DreamBank corpus [56]. They found that the word "run" in wakefulness reports occurred in a huge variety of contexts, including sports, means of transportation and programming, while in dream reports it resulted directly related to contexts referring to threatening events and chase/escape situations. The authors interpreted these observations as consistent with the so-called "Threat Simulation Theory", positing that dreams have an evolutionary function as a biological defense mechanism able to repeatedly simulate threatening events [70].

Sanz and colleagues used instead LSA to measure the degree of semantic relatedness between subjective reports produced by individuals under the effects of psychoactive substances and reports describing high or low lucidity dreams [71]. Dream reports were acquired through www.dreamjournal.net (accessed on 15 July 2021), a free service displaying over 200,000 dreams provided by more than 15,000 users [72]. Reports of psychoactive substance use, instead, were downloaded from the Erowid corpus [73]. The study evidenced that psychedelic lysergic acid diethylamide (LSD) elicited hallucinatory experiences more semantically similar to high lucidity dreams, while low lucidity dreams better resembled reports collected under the effect of deliriant tropane alkaloids. Moreover, the authors found that both in dream and hallucinogen reports, the most frequent words referred to perceptual modalities ("see", "visual", "face", "reality", "color"), emotions ("fear"), settings ("outside", "inside", "street", "front", "behind"), and relatives ("mom", "dad", "brother", "parent", "family").

Interestingly, word embedding methods were also applied within the above-cited study by Mota and colleagues [38] in order to investigate both the effect of the COVID-19 pandemic on the degree of incorporation into dreams of waking-life experiences, thoughts, and concerns related to the sanitary emergency. Rather than applying word embedding directly to the collected dataset, the authors trained a neural network based on the fastText model on a Wikipedia-derived corpus. Then, the obtained embeddings were used to calculate the distance between the words of dream reports and specific, preselected probe words

("contamination", "cleanness", "sickness", "health", "death", "life"). The study evidenced in pandemic dreams higher average semantic relatedness to the words "contamination" and "cleanness". Interestingly, the authors also found that pandemic-related semantic features of dream reports were related to the degree of participants' mental suffering (especially in terms of social withdrawal, isolation, cognitive impairment and recursive thinking), as estimated using the Positive and Negative Syndrome Scale (PANSS; [74]).

Pesonen and colleagues [75] also explored how the "new normality" subsequent to the outbreak of COVID-19 was incorporated into dreams and nightmares. The authors used crowdsourcing for collecting self-assessment of sleep quality and perceived level of stress, together with reports of dreaming experiences from 4275 respondents. An analysis of word associations was performed in order to verify the occurrence in dreams of thematic word categories related to the pandemic lockdown. The study detected 33 word clusters, of which 20 were evaluated as representing distressing content. The authors found an increased frequency of nightmares among participants who reported a higher level of stress during pandemic lockdown. Moreover, a subset of dreams was evaluated as highly pandemic-specific since it displayed word associations (e.g., handshake-distancing, mistake-hug, crowd-restriction) which were interpreted as referring to specific topics, such as "disregard of distancing" and "disease management".

### 3.4. Methodological Considerations

The studies described above showed that linguistic tools may allow for an automated and objective assessment of dream content from written or oral reports. Moreover, they do not require the use of blind raters, thus making the analysis less time and resource consuming and minimizing the impact of experimenter biases. However, it is important to specify here that these approaches also have important theoretical limitations, mainly concerning the inner ambiguity of language in general and more conspicuously of the language of dreams, often characterized by illogical or altered contexts and meanings. This implies that phenomena such as polysemy (i.e., the capacity for a word or phrase to have multiple meanings), homonymy (i.e., the relation between words with identical forms but different meanings), and contextual or metaphorical meanings may not always be handled properly by these techniques [76]. Moreover, the output of described analyses strongly depends on the chosen preprocessing approach. In fact, as described for graph analysis (Section 2.4), the dreamers' self-corrections and comments about the dream experience, may have to be removed manually. Hesitations, stutters, and word repetitions may be also removed. Moreover, typical preprocessing steps for distributional semantics approaches include: lowercase transformation, tokenization, lemmatization/stemming, removal of punctuation, non-alphabetic tokens, misspells, contractions and stop-words.

Importantly, dictionary-based and distributional semantics techniques such as word embedding do not differ only in what they can achieve, but also in their underlying assumptions and requirements. In fact, word embedding algorithms necessitate large amounts of data—possibly in the order of several million words—to produce highly stable and reliable results and, as such, are more computationally demanding. In this respect, it should be noted that free dream reports typically include ~100 words on average (although values may change considerably depending on several factors, such as the type of report or the sleep stage the oneiric experience was conceived in [24,27,76] and thus, several thousand or even tens of thousands reports may be necessary. Therefore, when relatively small datasets are analyzed, dictionary-based approaches may represent the only viable option. Alternatively, based on the specific study aims and hypotheses, one may consider computing the embeddings using an external, larger corpus, as in the recent work by Mota and colleagues (Section 3.3.2 [38]).

Concerning the use of word embedding methods, the choice of the embedding algorithm may also have crucial implications for data analysis and results interpretation. In this respect, the reader should be aware that the present review discussed only a few of the many available algorithms and approaches, which also have different advantages and

disadvantages with respect to each other. For instance, Altszyler and colleagues tested and compared LSA and word2vec approaches in the analysis of dream report collections of different sizes [67]. More specifically, the authors measured the performance of the models in determining the distance of the target word "run" with respect to escape/chase contexts. The study evidenced that LSA outperformed word2vec in smaller corpora (less than ~1 million words), being also able to capture word associations in collections with a low number of dreams and for low-frequency items. Other important differences between word embedding methods may concern, for example, their accuracy in the representation of low-frequency words or their computational demands.

## 4. Automated Methods for Replacement of Manual Ratings

In the previous sections, we described (semi-) automated techniques that may provide information about dream content similar or complementary to that obtained from canonical manual scoring, thanks in particular to the adoption of concepts and methods borrowed from the field of natural language processing. The same field could also provide us with the means for reproducing, in a fully automated fashion, the results of manual rating approaches such as the Hall and Van de Castle coding system [21,22]. This possibility was already illustrated by a series of studies by Bulkeley [57,59,60], who successfully identified some of the dream themes that are normally assessed through manual scoring by exploring dreams collected in dream databases using predefined sets of terms related to different categories of interest. More recently, Fogli and colleagues developed and validated a more advanced tool combining syntactic analysis with a dictionary-based approach [77]. In particular, the authors focused on three specific Hall and Van de Castle's categories ("characters", "interactions", and "emotions") and designed a dictionary-based tool that recognizes linguistic items recurring in dream reports as referring to specific entities of interest. In short, by applying a constituent-based analysis, the tool splits verbal data into constituents (i.e., group of words grammatically behaving as a single unit, such as phrasal categories or lexical categories) and, for each category of interest, intersects the set of constituents extracted from dream reports with a predefined list of linguistic units chosen as representative of a specific category (e.g., a database of verbs referring to aggression, friendliness, or sexual contacts for the category "interactions"). The authors validated this tool by verifying the association between waking-life and dreaming experiences on a corpus extracted from the DreamBank database. The tool not only allowed the authors to detect a certain degree of incorporation of waking-life features into dreams, but also showed the potential value of automated methods as a fast and reliable alternative to the manual annotation of dream reports. Crucially, however, the authors also noted that some themes included in the Hall and Van de Castle system (e.g., "success," "failure," "fortune," and "misfortune") might still be difficult to analyze using currently available natural language processing techniques, as they entail highly contextual and potentially ambiguous concepts.

## 5. Limitations and Future Challenges

As the above section exemplifies, while computational linguistics tools are increasingly applied in the field of dream research due to their capacity to provide similar or complementary information with respect to common manual approaches, their use still faces important limitations. However, such issues are being overcome or will likely be so within a few years, as available linguistics tools are continuously evolving, and novel approaches are being developed.

At present, one of the main challenges for automated methods is the evaluation of complex semantic contents, such as the identification of different kinds of interactions involving characters/objects, or higher-order meta-cognitive processes (e.g., self-awareness, counterfactual thinking, and theory of mind). This limitation may become evident especially when contents of interest are somewhat implicit or metaphorically expressed. However, it should be noted that even these cases could be adequately explored by approaches

based on dictionaries or distributional semantics, at least for particular contents, such as affective states. For instance, the sentence "As the monster approached, I began to sweat and tremble and tried to cry out but no sound would come", which implicitly refers to affective states and is scored as containing 'apprehension' (AP) based on the Hall and Van de Castle coding system [21], is assigned with three main emotional states by LIWC, which are fear (38.5%), sadness (38.5%), and disgust (23.1%). Similarly, a simple analysis based on word embeddings [78] akin to the one described by Mota and colleagues [38] shows that the above sentence is 'closer' to fear than to any other basic emotion. Therefore, while at present the assessment of human raters could represent a preferred option for the evaluation of complex semantic contents, future investigations should directly compare the reliability of manual and automated methods for different cases and features of interest. This comparison will inform the choice of the most appropriate approach according to the experimental aims of each study.

A more general limitation of dictionary-based and distributional semantics methods described in the present work is that they might not be able to adequately account for syntactically complicated descriptions, as in the case of negative sentences. Again, however, it should be noted that some approaches can already partially account for this issue when evaluating particular content, at least in simple cases. For example, LIWC assigns different relative scores of fear and disgust to the sentences "I saw a monster, and I was very scared" (fear = 85.9%; disgust = 14.1%) and "I saw a monster, and I was not scared" (fear = 40%; disgust = 60%). Moreover, other methods beyond those described in the present review, such as the recently introduced machine learning approach named 'Bidirectional Encoder Representations from Transformers' (BERT [79]), may be able to analyze complicated syntactic constructions efficiently. Indeed, BERT allows for a sophisticated analysis of texts by encoding sentence-level properties of linguistic units and exploring the hierarchical relationship between different words within the same syntactic context.

Another aspect that deserves to be discussed is the applicability of automated tools to different languages. Indeed, while all methods described within this review might be potentially applied to any language, the applicability and performance of each approach may be affected by the intrinsic characteristics of the language of interest and, at least in some cases, by the availability of adequate resources and tools. For instance, dictionary-based approaches such as LIWC or WordNet (cfr. Section 3.3.1), strongly depend on the presence of a large database (i.e., the dictionary) translated into the language of interest. For a precise analysis, this dictionary needs to be often updated, as the lexicon constantly changes across years—let's think, for instance, about words such as "covid-19" or "coronavirus", which in the last few years became part of our daily lexicon. In addition, because of intrinsic differences between languages and/or related tools, the methods described within this review typically need to be applied within the same language. Nonetheless, particular approaches might be applied to the specific aim of comparing how different languages represent specific meanings, such as in the case of emotion semantics [80].

Finally, a critical aspect of the application of automated tools on dream reports is the amount of data necessary for reliable and reproducible results (cfr. Section 3.4). In fact, a quantitative assessment of the number of dream reports or words per dream report needed for valid and well-grounded results is still lacking. Of note, this issue is especially relevant for the identification of regularities across oneiric experiences and the study of variables that may affect dream content due to the intrinsically large variability of experiences within and across individuals. In this respect, large-scale and data-sharing efforts will certainly play a key role in the coming years.

## 6. Conclusions

Dream research is interested in exploring aspects of dreaming conscious experience in quantifiable and comparable terms. To this aim, the verbal description provided by the sleeper upon awakening (i.e., the dream report) is the most viable and reliable proxy of an actual dream experience. Yet, the extrapolation of quantitative information from

such reports has always represented a central issue in the field of dream research. Starting from these considerations, the aim of this review was to outline an overall picture of the most recent approaches adopted to investigate dream activity from a linguistic point of view, with a particular focus on (semi-) automatic computational tools. In examining these methods, we presented their advantages, but we also evidenced their main limitations. Indeed, we showed that, on the one hand, automatic methods can produce results that are comparable to those of more classical manual approaches in a faster, fully reproducible, and objective fashion. These advantages are especially important for the analysis of large datasets, which is becoming more and more common (and necessary) in all fields of research. On the other hand, the described techniques may not be able at present to fully replace methods based on manual scoring systems, also due to a partial lack of adequate comparisons and validations. In addition, we showed, for automated tools, the importance of accurately planning and thoroughly describing both the preprocessing strategies applied and the reasons behind those methodological choices. In fact, to ensure replicability of results, researchers should always specify not only how the data was collected (e.g., oral vs. written reports) and analyzed overall, but also, and most importantly, how it was preprocessed. This should include, for oral reports, how the data was transcribed (e.g., automatically or manually, and verbatim or with changes/corrections). In this respect, our analysis of the literature also highlighted the necessity for further studies specifically investigating the impact of different methodological choices and computational strategies on final results. Finally, an implicit aim of this review was to encourage the sharing of data in order to overcome one of the main limitations of dream research in general and of research based on automated approaches in particular: the need for the collection of large amounts of data for the purpose of reliable and interpretable results. In fact, many of the methods that we described, and especially distributional semantics approaches, could express their full potential only if applied to very large datasets, including billions of words across hundreds of thousands of dream reports. We believe, indeed, that a collaborative effort by distinct sleep laboratories will represent a fundamental next step for deepening our knowledge and understanding of the origin and biological significance of dreams.

**Author Contributions:** Writing—original draft preparation, V.E. and G.B.; writing—review and editing, V.E., G.H. and G.B.; supervision, G.H. and G.B. All authors have read and agreed to the published version of the manuscript.

**Funding:** This work was supported by a BIAL Foundation Grant to G.B. and V.E., (#091/2020). G.B. is also supported by IBRO (Early Career Award 2019) and the European Research Council (ERC-2020-STG #948891).

**Institutional Review Board Statement:** Not applicable.

**Conflicts of Interest:** The authors declare no conflict of interest.

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
