# Peer review of "The Language of Dreams: Application of Linguistics-Based Approaches for the Automated Analysis of Dream Experiences"

_2624-5175, doi:10.3390/clockssleep3030035_

Round 1
Reviewer 1 Report
This is an excellent review of current research on dreams using digital tools of linguistic analysis. The authors focus on the various uses of graph analysis, which is interesting and important in itself. But the value of this paper involves its thorough consideration of the growing literature in this area. The following suggestions are meant as ways of strengthening an already strong paper. 1) Line 107, I’d say “enabling a more objective…” rather than “a fully objective…” 2) line 321 ff.: more of a cautionary note is needed here about over-screening dream reports to exclude what an external reader might consider “filler” or “nonsense,” but that might actually be meaningful expressions for the dreamer; in my own studies in this realm, I have to run the analyses with and without certain words to see if it matters; there may not be an a priori way to determine it. 3) line 558: It’s nice to see a reference to Schwartz 2004, one of the first to try this graphic analysis method, and an inspiration for many others who did later work in this area. Maybe more reference to her in the introduction? 4) line 654: Raising the issue of a need for large collections of dreams to use digital tools properly. This is a vital practical question in the field: how many dreams does one need to use a certain tool and get reliable results? This issue is mentioned again in the final lines, 732 ff. Perhaps more could be said about it earlier on, as an ongoing challenge for the field. 5) The paper does a great job identifying significant differences in the dreams of healthy people vs. psychiatric patients, and also addressing the methodological issue of long vs. short dreams. Somewhere, perhaps the introduction, discussion, and/or conclusion, comments could be added about the additional variables of different kinds of dreams (cf. Harry Hunt’s The Multiplicity of Dreams), different kinds of dreamers, and different kinds of settings in which dreams are shared (as anthropologists remind us), all of which can be relevant inputs for a digitally-driven analytic process.
Reviewer 2 Report
Overall, the paper is a very good introduction into the method of automated dream content analysis. I totally understand that the authors are fond of this approach, nevertheless, the limitation should be made clearer, and I also would suggest a paragraph of "How to do it."
- Dream research is interested in the dream experience, and the dream report is a more or less adequate proxy for this dream experience. This should be emphasized more clearly.
- The authors mention the HVC rating system but not other rating systems that are much easier to apply (that's somewhat biased, not the mention the alternatives).
- How would a automated analysis (or HVC in that case) code a dream report like "I saw a very big, blue monster and ran away as fast as I could." The dream report does not include a reference to emotion, but obviously the dream experience did. An human judge using the appropriate scales could code for that.
- Fine grained analysis. If one is interested in the interaction quality between dreamer and his/her partner in the dream, it's also not possible (so far) to use automated approaches, but sophisticated rating systems for social interactions are necessary. Another example are interaction with animals, is the dreamer caring for the animal, has fear etc. It should be discussed in the paper that some analyses require human judges.
- "methods may allow the extraction of quantitative information from language data, hence enabling a fully objective and reproducible analysis of conscious experiences occurring during human sleep." I don't agree with that as consicous experience is not equal to dream reports. A good example are the dream reports of schizophrenics, as they are told in waking life the thought disorder can be the factor affecting the dream report but not the dream experience. If I understand it correctly, waking reports and dream reports did not differ in that patient group.
- For a recent discussion of the pro and cons of oral vs. written dream reports see: Schredl, M., Dreer, J., Mösle, A., Rall, M., Rauch, L., Rose, S., & Seuffert, S. (2019). Voice-recorded vs. written dream reports: A research note. International Journal of Dream Research, 12(1), 138-140.
- Lastly, a section with a step-by-step instructions would be nice to have
1. How were the dreams collected (Pros and cons)
2. Preprocessing (I still think that need an human "eye".
3. What kind of research question should be answered, i.e., selection of the method (automated and/or human scoring) Pros and cons
Reviewer 3 Report
This manuscript contains a review of linguistic analytic approaches to studying dream content. I am familiar with the research literature on this topic and hope to provide some constructive feedback to the authors. Overall, the manuscript is clear and well-written. I recommend it for publication pending some minor revisions.
One thought I had while reading this review concerned the limitations of this linguistic analytic approach, which does not integrate well with script or storytelling approaches (Waters & Rodrigues-Doolabh, 2001; Waters & Waters, 2006) that have been used to study dreams in several studies (see Mikulincer et al., 2009; 2011; Selterman et al., 2012). The script-like analysis tradition has been used to assess personality dimensions such as attachment styles in rigorous ways. These studies have also found that 3rd person coders who score dreams for script-like content are able to deduce content which cannot be gathered through automated linguistic indicators because the sequence and structure of phrases/sentences has crucial significance beyond the mere presence or semantic meaning of the words themselves. The authors of the current review would do well to make note of this limitation to the linguistic approach.
Another limitation concerns higher order, meta-cognitive processes (e.g., self-awareness, counter-factual thinking, and theory of mind) which, to my knowledge, cannot be determined by the mere presence of words but must be deduced through perspective-taking coders.
Here are some more areas that I was hoping for more clarity about.
How might these approaches be applied to non-English languages?
How might these approaches help dreamers get more insight into themselves, in addition to the insights for researchers and clinicians? I understand some online dream repositories have also been utilized as quasi-social networks for users to share their dreams with others.
Overall I trust the authors to incorporate some minor changes and this will become a publishable manuscript.
